

# Exploiting nearest neighbor data and fuzzy membership function to address missing values in classification

Kurnia Muludi[1], Revita Setianingsih[2], Ridho Sholehurrohman[2] and Akmal Junaidi[2]

[1] Informatics and Business Institute Darmajaya, Bandar Lampung, Lampung Province, Indonesia
[2] Computer Science Department, Faculty of Science, Lampung University, Bandar Lampung, Lampung Province, Indonesia

## ABSTRACT

The accuracy of most classification methods is significantly affected by missing values. Therefore, this study aimed to propose a data imputation method to handle missing values through the application of nearest neighbor data and fuzzy membership function as well as to compare the results with standard methods. A total of five datasets related to classification problems obtained from the UCI Machine Learning Repository were used. The results showed that the proposed method had higher accuracy than standard imputation methods. Moreover, triangular method performed better than Gaussian fuzzy membership function. This showed that the combination of nearest neighbor data and fuzzy membership function was more effective in handling missing values and improving classification accuracy.

# INTRODUCTION

Complete and quality data are expected to be collected in the process of conducting any form of research (*Little & Rubin, 2019*). However, problems associated with data quality are observed to be increasing in the computer science field due to the introduction of data warehouse systems. This has led to the issue of poor data quality during the failure of projects in databases due to missing values and some other factors (*Tremblay & Hevner, 2021*).

Missing values are the situation where values or data points are absent in a dataset (*Kang, 2013*). This can be due to several reasons such as the lack of answers to some survey questions by respondents, inaccurate measurements and experimentation during manual entry of data, the existence of certain censored data, and other factors (*Kaiser, 2014*). There are three types of missing values which include Missing Completely at Random (MCAR), Missing at Random (MAR), and Not Missing at Random (NMAR) (*Emmanuel et al., 2021*). A thorough understanding of missing data nature and the careful selection of appropriate methods based on the assumptions and characteristics are important in solving this problem. However, the determination of the exact type can be challenging and

Corresponding author
Kurnia Muludi,
kurnia@darmajaya.ac.id

often relies on the understanding of the data, the context for the collection, and the statistical analysis applied.

This research was conducted to propose a simpler method of imputing missing values in five classification datasets by combining k-nearest neighbor (KNN) data points with fuzzy membership function. The method depended on the identification of closest observations with similar attributes to missing values. The value of 'k' for nearest neighbor was first determined followed by the calculation of the appropriate values based on the closest distance using the Euclidean distance formula. Subsequently, nearest neighbor value obtained was used as input into fuzzy membership formula to determine the weight. Missing values were imputed by calculating nearest value and weight using a weighted average function. The imputed data were evaluated in terms of accuracy using four classification algorithms and the results were compared to conventional imputation methods.

## RELATED WORKS

Missing values are normally handled using three different methods including deletion, imputation, and ensemble (*Emmanuel et al., 2021*). The first type of deletion method is pairwise which focuses on removing missing values and data from only calculated data pair while those in other features are preserved to allow the continuation of the analysis. The second type is list-wise which emphasizes removing all observations containing a minimum of one missing value. However, this method has the capacity to cause the loss of valuable information from the deleted data. The imputation method focuses on filing missing spaces with possible values based on known information. Moreover, ensemble is a method normally used to combine multiple models to produce a single better result (*Zhang et al., 2019*).

Machine learning-based imputation is an advanced method developed mainly to provide a predictive strategy to solve the problems of missing data using unsupervised or supervised learning. It has the capacity to estimate missing values using labeled or unlabeled data and based on the information obtained from available data. Moreover, imputation with high precision can be maintained for a longer period when the data provided have valuable information to address missing values. This has been achieved using several popular algorithms such as k-NN (*Taylor et al., 2022*), SVM (*Stewart, Zeng & Wu, 2018*), decision tree (*Rahman & Islam, 2013*), and clustering (*Zhang, Fang & Wang, 2016*).

The KNN algorithm has been widely used to solve missing value problems, specifically in datasets with more than one missing value feature (*Lin & Tsai, 2020*; *De Silva & Perera, 2016*; *Pujianto, Wibawa & Akbar, 2019*). The algorithm uses observations considered similar to others with missing values, also known as the target observation, in the process of imputing data. Moreover, the distance between these observations is normally measured through the Euclidean distance formula (*Witharana & Civco, 2014*). A significant weakness of KNN is the sensitivity to the distribution of missing data. This is due to the reliance of the algorithm on distance metrics which can be distorted to cause biased and

inaccurate results, low precision (*Beretta & Santaniello, 2016*), and higher computational time (*Acuna & Rodriguez, 2004*). Therefore, fuzzy logic was introduced to mitigate this weakness through the integration of a degree of uncertainty or membership to data points. Instead of treating missing values as completely absent or relying solely on traditional distance metrics, fuzzy logic assigns partial memberships to data points with due consideration for the similarity between the data points in a more nuanced manner.

Several cutting-edge methods have been developed for data imputation, each with specific advantages and disadvantages. For example, the MissForest method is effective at using random forests to handle mixed-type data even though the process is computationally demanding (*Stekhoven & Bühlmann, 2012*). KNN imputation is also a straightforward and intuitive method that preserves local patterns but the performance in high-dimensional spaces can be compromised by the number of neighbors selected (*Troyanskaya et al., 2001*). Moreover, SoftImpute is sensitive to regularization parameters and works effectively with noisy and high-dimensional datasets as well as using iterative soft-thresholding for matrix completion (*Mazumder, Hastie & Tibshirani, 2010*). Deep learning-based methods like the Generative Adversarial Imputation Network (GAIN) are also efficient at capturing intricate patterns but have several weaknesses such as the need for a substantial amount of data for training, proneness to overfitting specifically with limited data, and computationally demanding (*Yoon, Jordon & Schaar, 2018*).

The trend shows that fuzzy logic and KNN can be combined to provide a potentially effective way to overcome the drawbacks identified for each. Fuzzy logic offers a way to assign partial memberships to data points, allowing a more sophisticated study of the similarities compared to KNN associated with data distribution sensitivity problems. This combination can improve imputation accuracy in complex datasets relating to diseases such as hepatitis, blood transfusion, and Parkinson's speech. Therefore, this study successfully combined the local pattern preservation capabilities of KNN with a capacity of fuzzy logic to manage uncertainty in datasets where missing values possibly had a substantial influence on analysis results. The method was expected to reduce the difficulties associated with missing data and enable more thorough analysis in practical health applications.

# RESEARCH METHODOLOGY

## Proposed method

The material used in this study was the classification data retrieved from the UCI Machine Learning Repository with the percentage of missing values as presented in Table 1.

The method proposed for data imputation is presented in Fig. 1 and the first process was to determine KNN for each record without values for a specific property. This was followed by the application of fuzzy membership function to weigh missing feature values from the closest neighbors. The weighted mean was later used to calculate missing values. Moreover, triangular membership function was reported to have offered several benefits in determining the minimum, maximum, and average values as observed in the simplicity of

| No | Dataset name | Number of features | Number of observations | Missing data percentage |
|----|--------------|--------------------|-----------------------|-------------------------|
| 1 | Hepatitis | 20 | 155 | 5.39% |
| 2 | Ozone level detection | 73 | 2.534 | 8.07% |
| 3 | Blood transfusion | 5 | 748 | 5% and 10% |
| 4 | Parkinson speech | 26 | 1.040 | 5% and 10% |
| 5 | Audit risk | 26 | 776 | 5% and 10% |

Table 1 Details of research materials.

use, improved convenience, quick calculations, and quick responses (*Abdel-Basset, Mohamed & Chang, 2018*; *Roman, Precup & Petriu, 2021*). The average value and standard deviation were computed to be used as input in Gaussian function.

Algorithm 1 provides details of the proposed data imputation algorithm using KNN data and the triangle membership function while Algorithm 2 has comprehensive information on Gaussian aspect. Moreover, triangular membership function was used to describe fuzzy set with triangular shape and characterized by three parameters including the starting point (a), the peak point (b), and the endpoint (c) (*Azam et al., 2020*) presented as follows:

$$\mu(x) = \begin{cases} 0 & x \leq a \\ \frac{x-a}{b-a} & a < x \leq b \\ \frac{c-x}{c-b} & b < x < c \\ 0 & x \geq c \end{cases} \quad (1)$$

Gaussian function was used to describe fuzzy set with the shape of Gaussian curve or bell curve. The process was based on two parameters which were the mean or center of the curve ($\mu$) and the standard deviation or curve width ($\sigma$) (*Azam et al., 2020*) presented as follows:

$$\mu(x) = e^{-\left(\frac{(x-\mu)^2}{2\sigma^2}\right)} \quad (2)$$

## Method evaluation

Performance evaluation was conducted to determine the effectiveness of the proposed method when compared to traditional imputation methods. This was achieved using five datasets with different missing value percentages and the metric used for comparison was the accuracy. The last part in Fig. 1 shows the stages of data classification using KNN, Naive Bayes, Decision Tree, and Neural Network, along with the respective accuracy scores while the final stage focuses on the comparison. It is important to state that the confusion matrix is one of the popular tools normally used to evaluate classification performance (*Ruuska et al., 2018*). This matrix contains information about the classification prediction results compared to the actual data generated by the system applied. The results are often presented in the form of four combinations including true positive (TP), true negative (TN), false positive (FP), and false negative (FN) (Table 2).

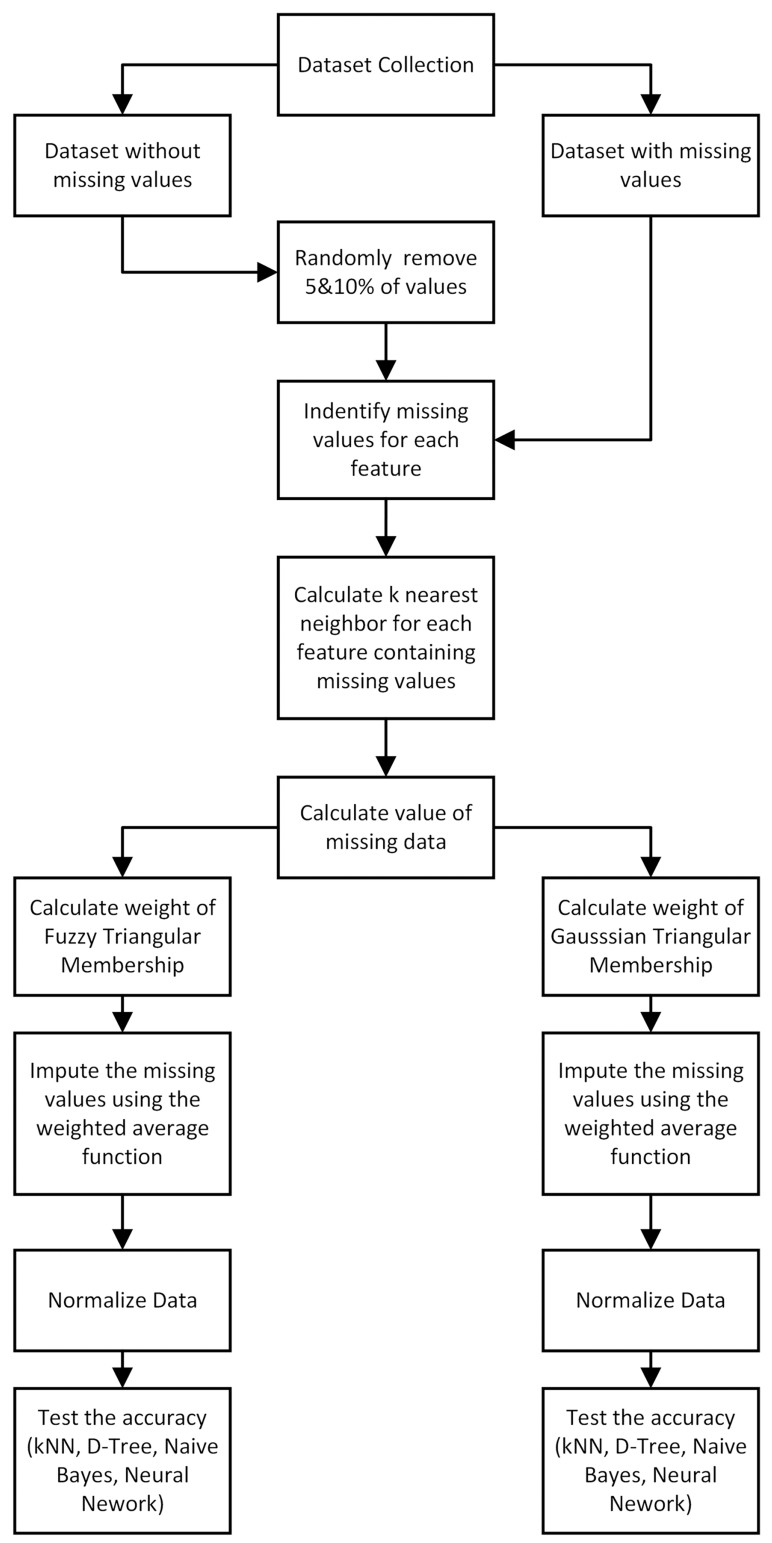

**Figure 1** **The proposed method of data imputation.**

---

**Algorithm 1** Data imputation with k nearest neighbour data and triangular fuzzy.

1: Determine $k$, which is the number of closest observations.

2: Calculate the closest distance between the target observation and observations that do not have missing values.

3: Find the closest observation value with the minimum distance.

4: Calculate the minimum value (nearest neighbor observation value).

5: Calculate the maximum value (nearest neighbor observation value).

6: Calculate the average value (nearest neighbor observation value).

7: Calculate the weight of the value using the triangular fuzzy membership function.

8: Calculate the weighted average value of the nearest neighbor weights and values.

$$\frac{\sum_{x \in X} \text{Nearest Neighbor Values}(x) \times \text{Triangular weight}(x)}{\sum_{x \in X} \text{Triangular weight}(x)}$$

9: End.

---

**Algorithm 2** Data imputation with k nearest neighbour data and Gaussian fuzzy.

1: Determine $k$, which is the number of closest observations.

2: Calculate the closest distance between the target observation and observations that do not have missing values.

3: Find the closest observation value with the minimum distance.

4: Calculating the average value (nearest neighbor observation value).

5: Calculate the standard deviation (nearest neighbor observation value).

6: Calculate the weight of the value using the membership function of Gaussian fuzzy.

7: Calculate the weighted average value of the nearest neighbor weights and values.

$$\frac{\sum_{x \in X} \text{Nearest Neighbor Values}(x) \times \text{Gaussian weight}(x)}{\sum_{x \in X} \text{Gaussian weight}(x)}$$

8: End.

---

**Table 2 Confusion matrix.**

| Class | Predictive positive | Predictive negative |
|---|---|---|
| Actual positive | TP | FN |
| Actual negative | FP | TN |

The accuracy was calculated to estimate the precision level of the classification results on the data (*Deng et al., 2016*). This was achieved using the following formula (*Pradana & Hayaty, 2019*).

$$\text{Accuracy} = \left( \frac{TP + TN}{TP + TN + FP + FN} \right) \times 100 \qquad (3)$$

| A | B | C | D | E | F |
|---|---|---|---|---|---|
| 2 |  | 100 | 0.2 | 2 | 0 |
| 3 | 51 | 400 | 0.7 |  | 1 |
| x | 23 | 100 | 0.9 | 3 |  |
| 7 |  | 200 | 0.8 | 5 | 1 |
| 1 | 11 |  | 0.1 | 4 | 0 |

**Figure 2 Illustration of data for closest value calculation on x.**

## RESULTS AND DISCUSSION

The two imputation methods proposed in this study were compared with conventional methods based on classification performance accuracy to determine the effectiveness. K-neighbor data were selected and applied through the identification of missing values index followed by the determination of the closest distance between missing values and target observations using the data presented in Fig. 2.

The small dataset example in the figure was used to explain the selection of nearest neighbor data points. Suppose the aim was to determine the distance from missing values index located in column A of the third row, Columns B, C, D, E, and F in the third row would be compared with the corresponding columns in the first to last rows. The determination of these distances would be followed by the selection of the closest or smallest to the calculation results. Assuming 'k' was set to 3, three values would be retrieved from column A, corresponding to the rows with the smallest distances, and considered the closest values. An example of the method to calculate the distance is presented as follows:

- Calculate the distance between row 3 and 1 of col A:

$$d(3,1) = \sqrt{\sum_{i=1}^{5} (x_3^i - x_1^i)^2}$$

$$= \sqrt{(23-0)^2 + (100-100)^2 + (0.9-0.2)^2 + (3-2)^2 + (0-0)^2}$$

$$= 23.03237$$

- Calculate the distance between row 3 and 2 of col A:

$$(3,2) = \sqrt{\sum_{i=1}^{5} (x_3^i - x_2^i)^2}$$

$$= \sqrt{(23-51)^2 + (100-400)^2 + (0.9-0.7)^2 + (3-0)^2 + (0-1)^2}$$

$$= 301.3205$$

| A | B | C | D | E | F | d(x,y) |
|---|---|---|---|---|---|---|
| 2 | | 100 | 0.2 | 2 | 0 | 23.0324 |
| 3 | 51 | 400 | 0.7 | | 1 | 301.321 |
| x | 23 | 100 | 0.9 | 3 | | - |
| 7 | | 200 | 0.8 | 5 | 1 | 102.635 |
| 1 | 11 | | 0.1 | 4 | 0 | 100.726 |

**Figure 3  Illustration of the nearest value.**     

- Calculate the distance between row 3 and 4 of col A:

$$d(3,4) = \sqrt{\sum_{i=1}^{5}(x_3^i - x_4^i)^2}$$
$$= \sqrt{(23-0)^2 + (100-200)^2 + (0.9-0.8)^2 + (3-5)^2 + (0-1)^2}$$
$$= 102.6353$$

- Calculate the distance between row 3 and 5 of col A:

$$d(3,5) = \sqrt{\sum_{i=1}^{5}(x_3^i - x_5^i)^2}$$
$$= \sqrt{(23-11)^2 + (100-0)^2 + (0.9-0.1)^2 + (3-4)^2 + (0-0)^2}$$
$$= 100.7256$$

The calculation of the distance for each row showed that the three closest values were 2, 1, and 7 followed by the determination of the respective weights using fuzzy membership function. The results obtained from the calculation of the distance are presented in Fig. 3 to determine the closest or nearest value.

The calculation processes for both triangular fuzzy and Gaussian functions were the same except for the differences in the formula for each. Therefore, the determination of nearest value is based on the closest distance calculated. This was followed by a weight search using nearest value obtained as the input in triangular fuzzy and Gaussian functions. The weight ranged from 0 to 1 and the figures obtained for the closest values of 2, 1, and 7 using triangular fuzzy membership are stated as follows:

- Calculate value weight for nearest value = 2

$$\mu(2) = \frac{x-a}{b-a}$$
$$= \frac{2-1}{3.33-1}$$
$$= 0.428571$$

| A | B | C | D | E | F |
|---|---|---|---|---|---|
| 2 |  | 100 | 0.2 | 2 | 0 |
| 3 | 51 | 400 | 0.7 |  | 1 |
| 2 | 23 | 100 | 0.9 | 3 |  |
| 7 |  | 200 | 0.8 | 5 | 1 |
| 1 | 11 |  | 0.1 | 4 | 0 |

**Figure 4 Illustration of imputation with triangular fuzzy.**

- Calculate value weight for nearest value = 1

$$\mu(1) = 0$$

- Calculate value weight for nearest value = 7

$$\mu(7) = 0$$

The weight of triangular fuzzy membership function was determined by identifying the minimum, maximum, and average values from the closest data points. Both the minimum and maximum were assigned a membership weight of 0 and the values in the range were subsequently used in the membership function calculation. This was achieved by using the weights obtained as inputs in the weighted average formula. The process focused on multiplying the sum of values with the respective weights and dividing the result by the total sum of weights. The resulting value was later used as an imputation for the index that previously contained missing data. The imputation process is presented in Fig. 4 to show the weighted average value calculated from nearest data points and associated weights using triangular fuzzy membership function.

$$
\begin{aligned}
\text{Weighted Value} &= \frac{\sum_{x \in X} \text{Nearest Neighbor Values}(x) \times \text{Triangular weight}(x)}{\sum_{x \in X} \text{Triangular weight}(x)} \\
&= \frac{(2 \times 0.428571) + (1 \times 0) + (7 \times 0)}{0.428571 + 0 + 0} \\
&= \frac{2 \times 0.428571}{0.428571} \\
&= 2
\end{aligned}
$$

The weights for Gaussian fuzzy membership function were obtained by determining the average and standard deviation of the closest values and used as input in the weighted average formula for imputing missing values in indexes. The example of the methods used in calculating the weight for closest values 2, 1, and 7 is presented as follows. Moreover, imputation using the weighted average value derived from Gaussian fuzzy membership function is presented in Fig. 5.

| A | B | C | D | E | F |
|---|---|---|---|---|---|
| 2 |  | 100 | 0.2 | 2 | 0 |
| 3 | 51 | 400 | 0.7 |  | 1 |
| 2.63 | 23 | 100 | 0.9 | 3 |  |
| 7 |  | 200 | 0.8 | 5 | 1 |
| 1 | 11 |  | 0.1 | 4 | 0 |

**Figure 5 Illustration of the nearest value.**

• Calculate value weight for nearest value = 2

$$\mu(2) = e^{-\frac{(2-\mu)^2}{2\sigma^2}}$$
$$= 2.718^{-(2-3.33)^2/2(0.7698)^2}$$
$$= 0.878957$$

• Calculate value weight for nearest value = 1

$$\mu(1) = e^{-\frac{(1-\mu)^2}{2\sigma^2}}$$
$$= 2.718^{-(1-3.33)^2/2(0.7698)^2}$$
$$= 0.673599$$

• Calculate value weight for nearest value = 7

$$\mu(7) = e^{-\frac{(7-\mu)^2}{2\sigma^2}}$$
$$= 2.718^{-(7-3.33)^2/2(0.7698)^2}$$
$$= 0.376926$$

$$\text{Weighted Value} = \frac{\sum_{x\in X} \text{Nearest Neighbor Values}(x) \times \text{Gaussian weight}(x)}{\sum_{x\in X} \text{Gaussian weight}(x)}$$
$$= \frac{(2 \times 0.878957) + (1 \times 0.673599) + (7 \times 0.376926)}{0.878957 + 0.376926 + 0.673599}$$
$$= \frac{5.069999}{1.929483}$$
$$= 2.627647$$
$$= 2.63$$

The proposed method with KNN classifier was evaluated by setting the number of nearest neighbors for imputation at 3, 5, 7, and 9. Further classification was subsequently conducted by partitioning the data into training and test sets followed by a normalization process using the Min-Max method (*Henderi, Wahyuningsih & Rahwanto, 2021*). Moreover, KNN Classifier function was configured with Euclidean metrics to measure data proximity. The Naive Bayes model was constructed using Gaussian function while the Neural Network model used the multi-layer perceptron function, and both were applied to

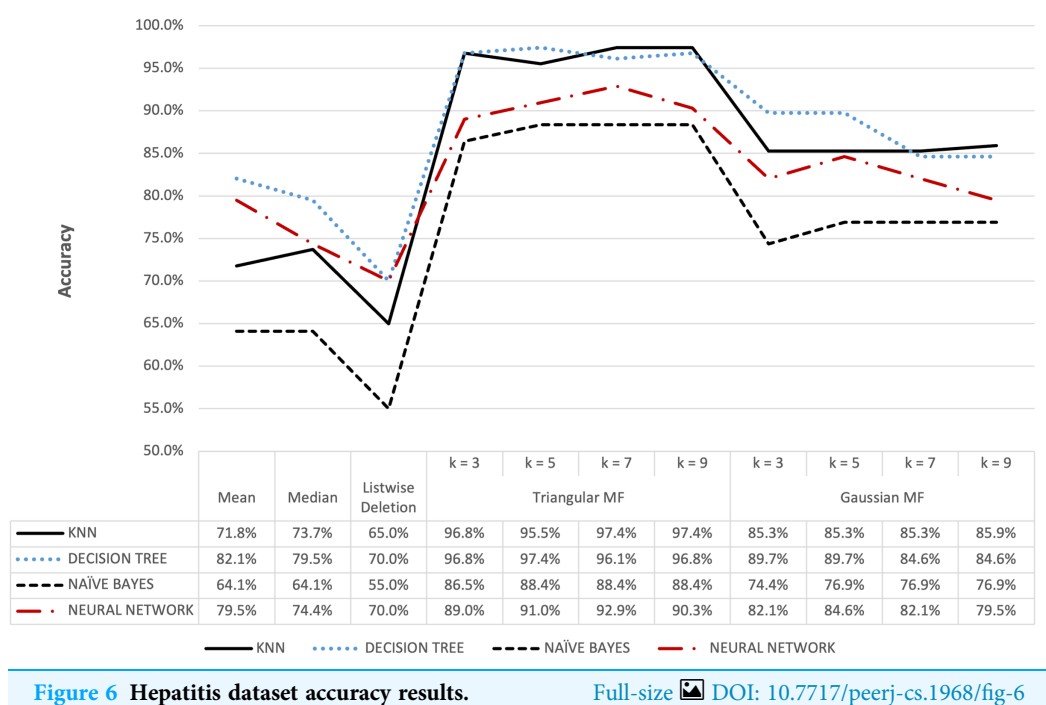

**Figure 6 Hepatitis dataset accuracy results.**

make predictions on the test data. The completion of the imputation and classification stages for the five datasets was followed by the evaluation of the performance of the models, precisely the accuracy, using the confusion matrix, and the results are presented in Figs. 6–13.

Figure 6 shows that triangular and Gaussian fuzzy membership functions have higher accuracy values compared to mean and median imputation as well as listwise deletion methods. This is clearly observed from the upper accuracy range of triangular membership function for k = 7 and k = 9 with KNN accuracy of 97.44% compared to 82.05% recorded for mean imputation using decision tree. Moreover, the comparison of the performance accuracy in Fig. 7 showed that imputation using fuzzy membership function was better than the conventional methods. This was confirmed by the maximum accuracy value of 97.87% obtained with triangular fuzzy membership function and k = 7 in KNN classification, thereby showing the ability to improve classification accuracy results.

Figures 8 and 9 further show that fuzzy membership function is superior in terms of accuracy in classifying missing data, 5% or 10%, compared to conventional methods. This showed that fuzzy membership function consistently provided an accurate classification for missing data, regardless of the percentage being small or large. The trend was confirmed by the maximum accuracy value recorded for triangular to be 83.96% for k = 5 while Gaussian was 86.10% for k = 3 with data containing 10% missing data.

The accuracy for the Parkinson speech dataset containing 5% and 10% missing data is presented in Figs. 10 and 11. The results were similar to the blood transfusion dataset where fuzzy membership function had better accuracy for both 5% and 10% missing data compared to the conventional methods. The maximum value of 97.69% was recorded for the imputation using triangular membership function at k = 5 while Gaussian had 96.54%

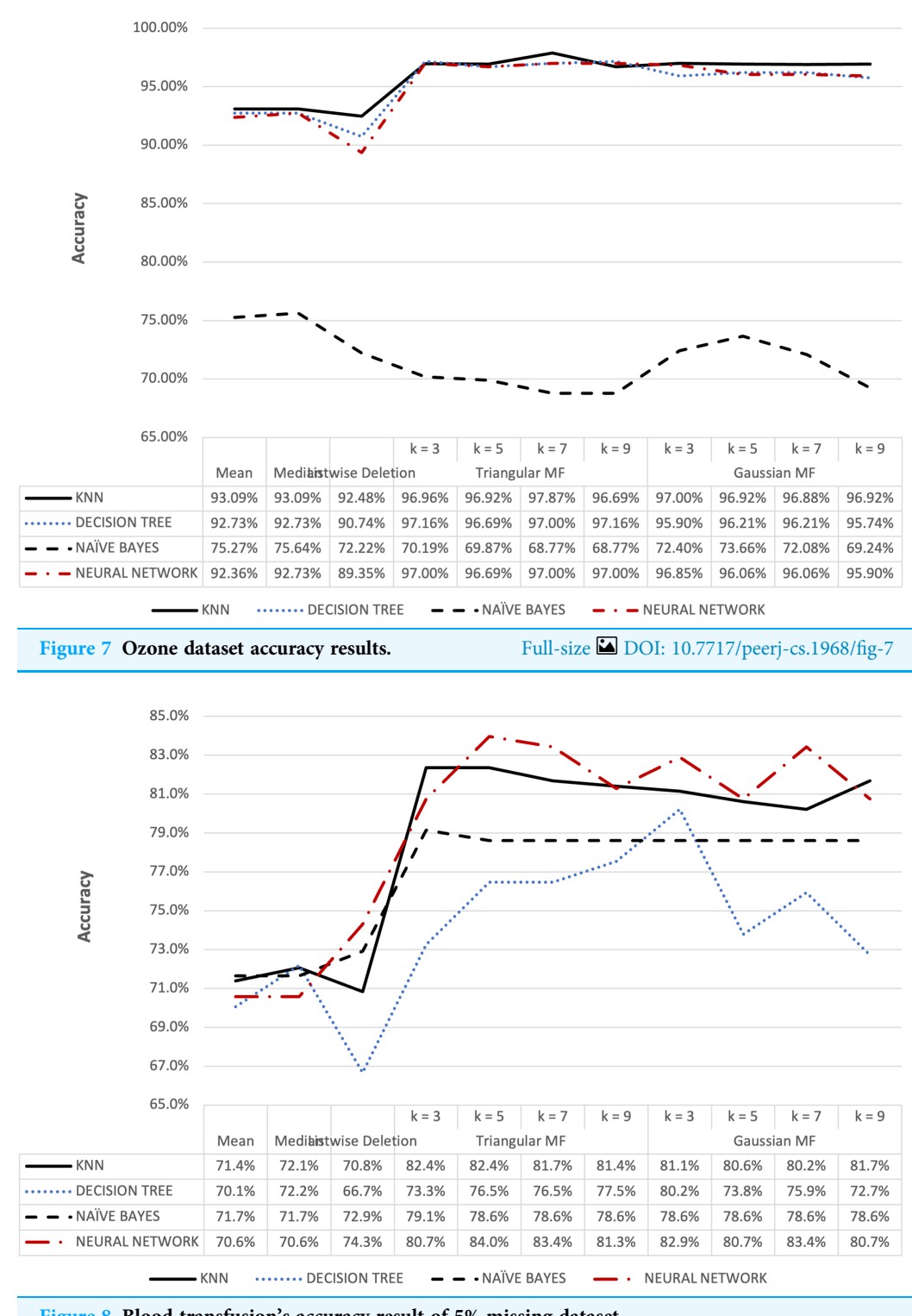

**Figure 7  Ozone dataset accuracy results.**     

|  | Mean | Median | Listwise Deletion | Triangular MF | | | | Gaussian MF | | | |
|---|---|---|---|---|---|---|---|---|---|---|---|
|  |  |  |  | k = 3 | k = 5 | k = 7 | k = 9 | k = 3 | k = 5 | k = 7 | k = 9 |
| KNN | 93.09% | 93.09% | 92.48% | 96.96% | 96.92% | 97.87% | 96.69% | 97.00% | 96.92% | 96.88% | 96.92% |
| DECISION TREE | 92.73% | 92.73% | 90.74% | 97.16% | 96.69% | 97.00% | 97.16% | 95.90% | 96.21% | 96.21% | 95.74% |
| NAÏVE BAYES | 75.27% | 75.64% | 72.22% | 70.19% | 69.87% | 68.77% | 68.77% | 72.40% | 73.66% | 72.08% | 69.24% |
| NEURAL NETWORK | 92.36% | 92.73% | 89.35% | 97.00% | 96.69% | 97.00% | 97.00% | 96.85% | 96.06% | 96.06% | 95.90% |

**Figure 8  Blood transfusion's accuracy result of 5% missing dataset.**

|  | Mean | Median | Listwise Deletion | Triangular MF | | | | Gaussian MF | | | |
|---|---|---|---|---|---|---|---|---|---|---|---|
|  |  |  |  | k = 3 | k = 5 | k = 7 | k = 9 | k = 3 | k = 5 | k = 7 | k = 9 |
| KNN | 71.4% | 72.1% | 70.8% | 82.4% | 82.4% | 81.7% | 81.4% | 81.1% | 80.6% | 80.2% | 81.7% |
| DECISION TREE | 70.1% | 72.2% | 66.7% | 73.3% | 76.5% | 76.5% | 77.5% | 80.2% | 73.8% | 75.9% | 72.7% |
| NAÏVE BAYES | 71.7% | 71.7% | 72.9% | 79.1% | 78.6% | 78.6% | 78.6% | 78.6% | 78.6% | 78.6% | 78.6% |
| NEURAL NETWORK | 70.6% | 70.6% | 74.3% | 80.7% | 84.0% | 83.4% | 81.3% | 82.9% | 80.7% | 83.4% | 80.7% |

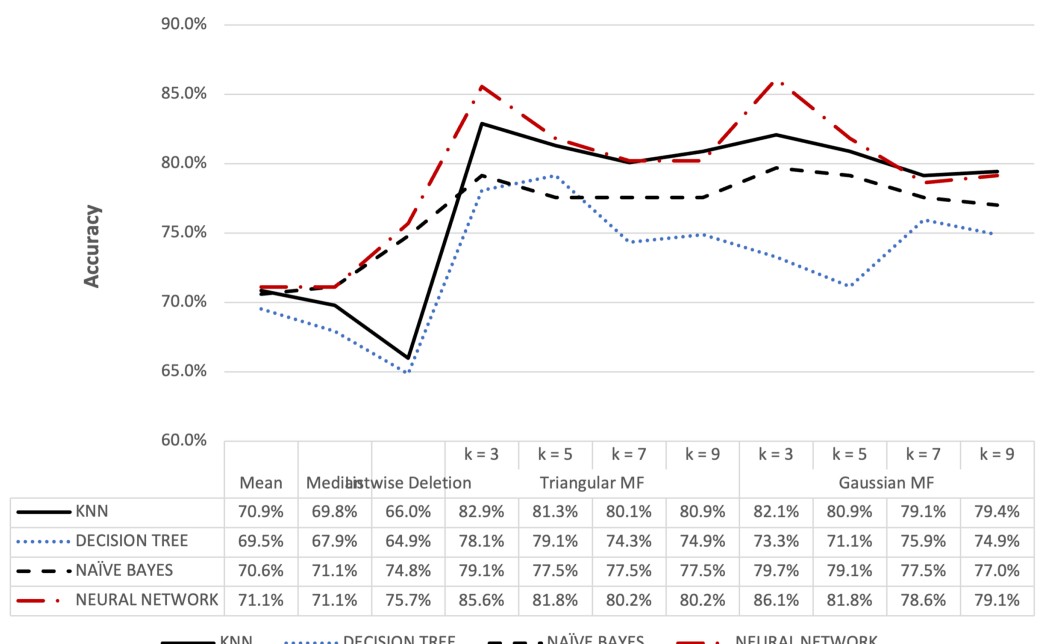

**Figure 9 Blood transfusion's accuracy result of 10% missing dataset.**

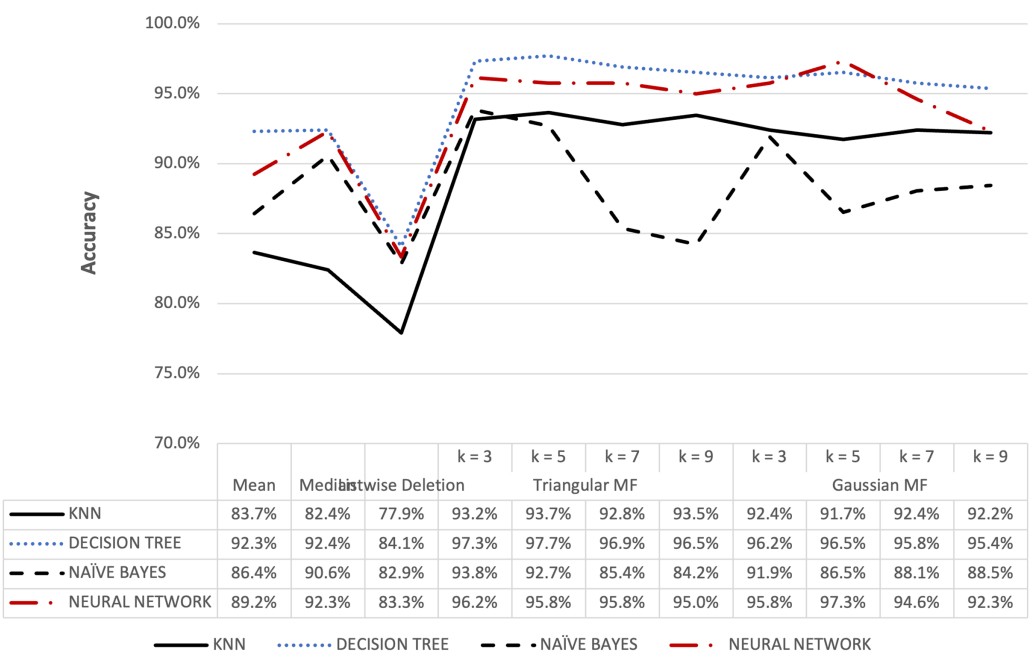

**Figure 10 Parkinson's accuracy result of 5% missing dataset.**

at k = 3. This showed that the imputation process with fuzzy membership function could be applied to optimally enhance accuracy outcomes compared to conventional methods.

Figures 12 and 13 show the accuracy of the audit risk datasets and fuzzy membership function were observed to have better results compared to conventional methods. For

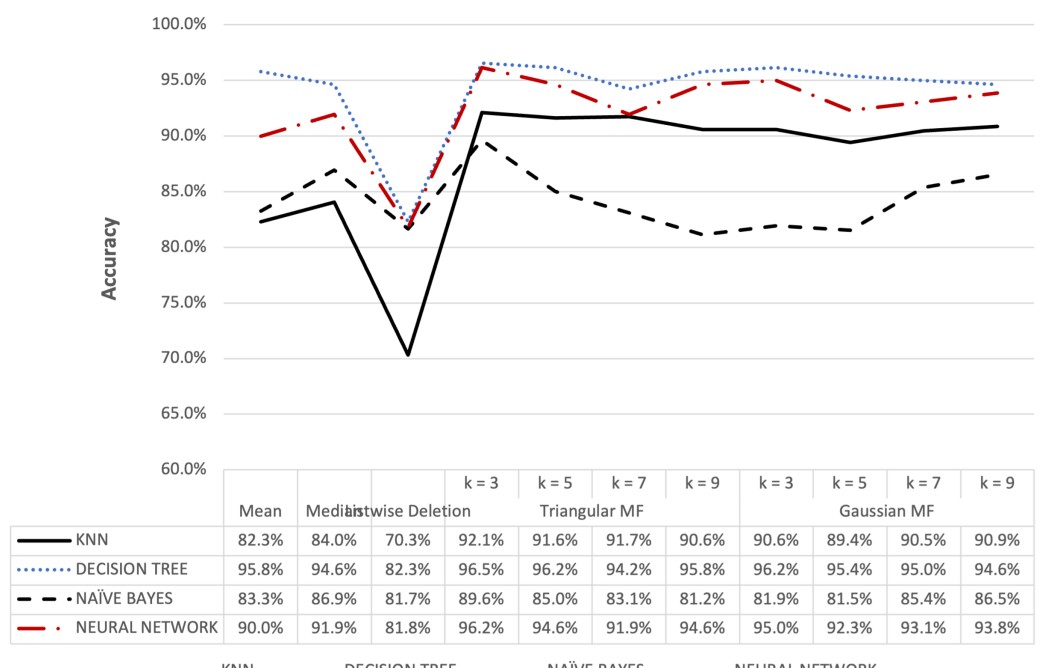

| | Mean | Median | Listwise Deletion | k = 3 | k = 5 | k = 7 | k = 9 | k = 3 | k = 5 | k = 7 | k = 9 |
|---|---|---|---|---|---|---|---|---|---|---|---|
| | | | | Triangular MF | | | | Gaussian MF | | | |
| KNN | 82.3% | 84.0% | 70.3% | 92.1% | 91.6% | 91.7% | 90.6% | 90.6% | 89.4% | 90.5% | 90.9% |
| DECISION TREE | 95.8% | 94.6% | 82.3% | 96.5% | 96.2% | 94.2% | 95.8% | 96.2% | 95.4% | 95.0% | 94.6% |
| NAÏVE BAYES | 83.3% | 86.9% | 81.7% | 89.6% | 85.0% | 83.1% | 81.2% | 81.9% | 81.5% | 85.4% | 86.5% |
| NEURAL NETWORK | 90.0% | 91.9% | 81.8% | 96.2% | 94.6% | 91.9% | 94.6% | 95.0% | 92.3% | 93.1% | 93.8% |

KNN ......... DECISION TREE — — · NAÏVE BAYES — · NEURAL NETWORK

**Figure 11 Parkinson's accuracy result of 10% missing dataset.**

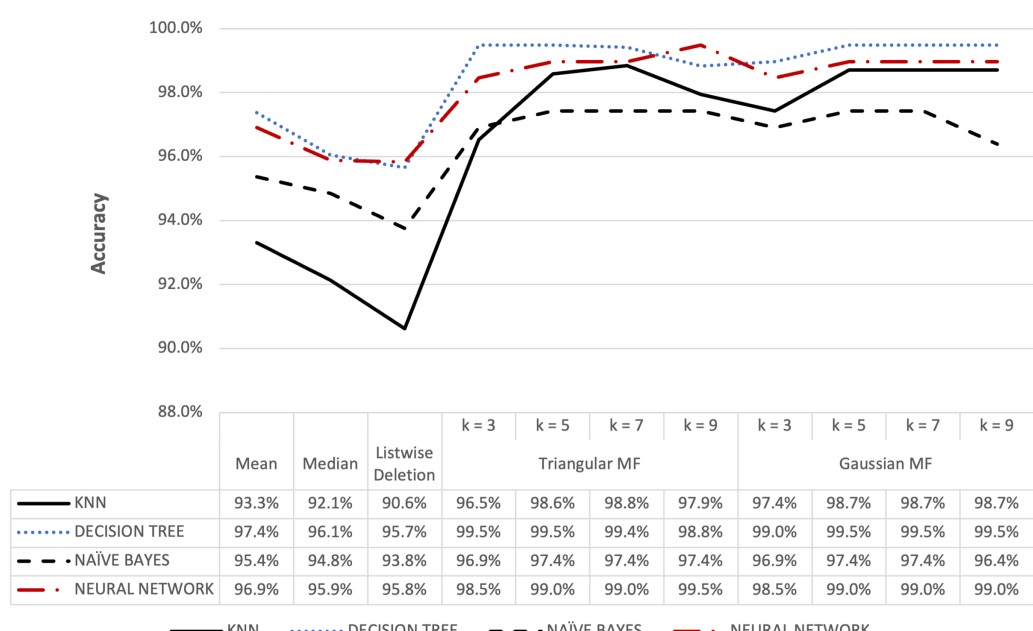

| | Mean | Median | Listwise Deletion | k = 3 | k = 5 | k = 7 | k = 9 | k = 3 | k = 5 | k = 7 | k = 9 |
|---|---|---|---|---|---|---|---|---|---|---|---|
| | | | | Triangular MF | | | | Gaussian MF | | | |
| KNN | 93.3% | 92.1% | 90.6% | 96.5% | 98.6% | 98.8% | 97.9% | 97.4% | 98.7% | 98.7% | 98.7% |
| DECISION TREE | 97.4% | 96.1% | 95.7% | 99.5% | 99.5% | 99.4% | 98.8% | 99.0% | 99.5% | 99.5% | 99.5% |
| NAÏVE BAYES | 95.4% | 94.8% | 93.8% | 96.9% | 97.4% | 97.4% | 97.4% | 96.9% | 97.4% | 97.4% | 96.4% |
| NEURAL NETWORK | 96.9% | 95.9% | 95.8% | 98.5% | 99.0% | 99.0% | 99.5% | 98.5% | 99.0% | 99.0% | 99.0% |

KNN ......... DECISION TREE — — · NAÏVE BAYES — · NEURAL NETWORK

**Figure 12 Audit's accuracy result of 5% missing dataset.**

missing value percentages of 5% and 10%, the maximum accuracy obtained for both triangular and Gaussian functions was 99.48%. These results were consistent with the observation in other datasets where fuzzy membership function was more optimal in
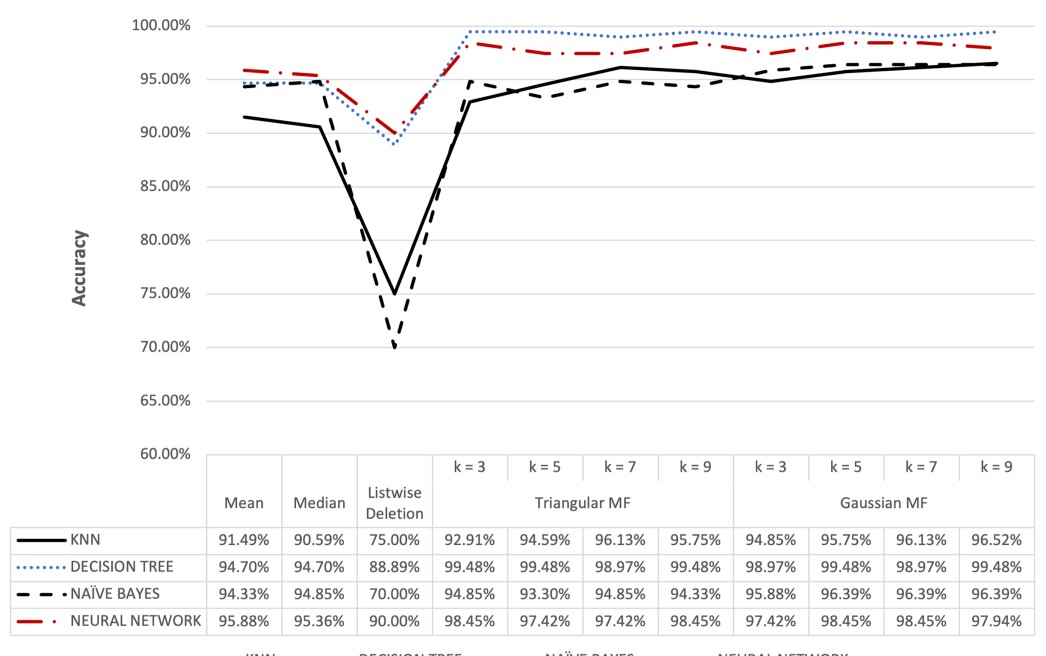

| | Mean | Median | Listwise Deletion | k = 3 | k = 5 | k = 7 | k = 9 | k = 3 | k = 5 | k = 7 | k = 9 |
| --- | --- | --- | --- | --- | --- | --- | --- | --- | --- | --- | --- |
| | | | | Triangular MF | | | | Gaussian MF | | | |
| KNN | 91.49% | 90.59% | 75.00% | 92.91% | 94.59% | 96.13% | 95.75% | 94.85% | 95.75% | 96.13% | 96.52% |
| DECISION TREE | 94.70% | 94.70% | 88.89% | 99.48% | 99.48% | 98.97% | 99.48% | 98.97% | 99.48% | 98.97% | 99.48% |
| NAÏVE BAYES | 94.33% | 94.85% | 70.00% | 94.85% | 93.30% | 94.85% | 94.33% | 95.88% | 96.39% | 96.39% | 96.39% |
| NEURAL NETWORK | 95.88% | 95.36% | 90.00% | 98.45% | 97.42% | 97.42% | 98.45% | 97.42% | 98.45% | 98.45% | 97.94% |

**Figure 13 Audit's accuracy result of 10% missing dataset.**

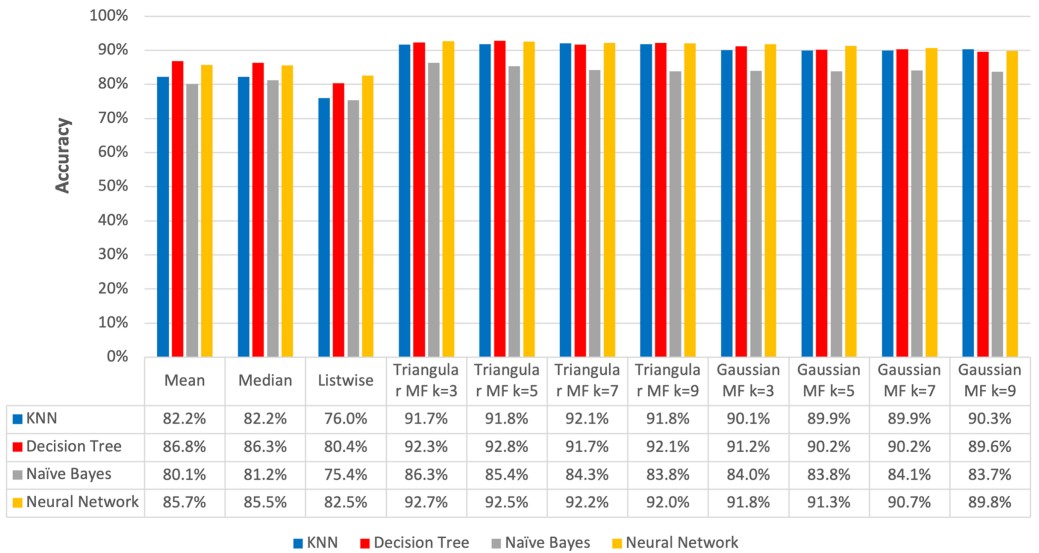

| | Mean | Median | Listwise | Triangular MF k=3 | Triangular MF k=5 | Triangular MF k=7 | Triangular MF k=9 | Gaussian MF k=3 | Gaussian MF k=5 | Gaussian MF k=7 | Gaussian MF k=9 |
| --- | --- | --- | --- | --- | --- | --- | --- | --- | --- | --- | --- |
| KNN | 82.2% | 82.2% | 76.0% | 91.7% | 91.8% | 92.1% | 91.8% | 90.1% | 89.9% | 89.9% | 90.3% |
| Decision Tree | 86.8% | 86.3% | 80.4% | 92.3% | 92.8% | 91.7% | 92.1% | 91.2% | 90.2% | 90.2% | 89.6% |
| Naïve Bayes | 80.1% | 81.2% | 75.4% | 86.3% | 85.4% | 84.3% | 83.8% | 84.0% | 83.8% | 84.1% | 83.7% |
| Neural Network | 85.7% | 85.5% | 82.5% | 92.7% | 92.5% | 92.2% | 92.0% | 91.8% | 91.3% | 90.7% | 89.8% |

**Figure 14 Average accuracy of imputation methods.**

improving accuracy during the process of handling missing values compared to conventional methods.

## Comparison of accuracy results

The accuracy results of the methods applied were compared to determine the best in resolving the issues of missing values. This was achieved using five different classification datasets and the average accuracy results presented in Fig. 14 showed that the imputation

using fuzzy membership function had a higher average accuracy compared to standard or conventional imputation. The results showed that the standard methods treated missing values as either completely absent or imputed through a single fixed value, leading to the oversimplification of the reality associated with data uncertainty. Meanwhile, fuzzy logic allowed the assignment of degrees of membership to different potential values for missing data in order to reflect the uncertainty.

The average accuracy of the imputation using triangular fuzzy membership function ranged from 91.72% to 92.07% with the maximum recorded at k = 7 in KNN classification method while the Decision Tree method was from 91.68% to 92.82% and the highest was at k = 5. Moreover, Naive Bayes method was from 83.81% to 86.27% with the maximum at k = 3 and Neural Network had 92.05% to 92.70% with the highest also found at k = 3.

Gaussian function produced average accuracy ranging from 89.90% to 90.28% and the maximum was observed at k = 9 in KNN classification. Meanwhile, the Decision Tree method produced from 89.61% to 91.17% and the highest recorded at k = 3 imputation, Naive Bayes ranged from 83.70% to 84.05% with a peak found at k = 7, and Neural Network classification was from 89.79% to 91.82% with the maximum identified at k = 3.

The highest average accuracy among the five imputation methods was 92.82% using triangular fuzzy membership function with k = 5 based on a Decision Tree classification, showing the method was the best to overcome the problem of missing values. This was in line with the results of a previous study conducted by *El-Bakry et al. (2021)* on the same dataset, specifically the ozone level detection dataset, that using the triangular fuzzy membership function with a Decision Tree classification yielded an accuracy of 96%. Meanwhile, the Gaussian fuzzy membership function achieved an accuracy of 92%. When employing a neural network for classification, the accuracy using the triangular fuzzy membership function was 90% while the Gaussian fuzzy membership function was 88%.

Finally, the classification using Naive Bayes resulted in an accuracy of 73% when utilizing the triangular method and 69% when using the Gaussian method. For evaluation using KNN classification method, highest accuracy value was obtained with the triangular at 98.11% and the Gaussian was 97.48%. The classification using Decision Tree provided the maximum accuracy of 97.16% for the triangular and 96.21% for the Gaussian. The classification using Naive Bayes reached a maximum accuracy of 70.19% for the triangular and 73.66% for the Gaussian. The classification using Neural Network provided a maximum accuracy of 97% for the triangular and 96.85% Gaussian membership. Figure 15 further compares the results of the present and previous studies on the ozone level detection dataset. Based on the discussion, the accuracy results of this study exceed the maximum accuracy results achieved by previous studies for each method and the improvement was attributed to the prepossessing.

The results showed that the accuracy obtained in this study exceeded the maximum achieved by previous studies for each method. This improvement was attributed to the preprocessing step of normalizing the training and test data as described in *Henderi, Wahyuningsih & Rahwanto (2021)* to re-scale and eliminate variations in feature scales in

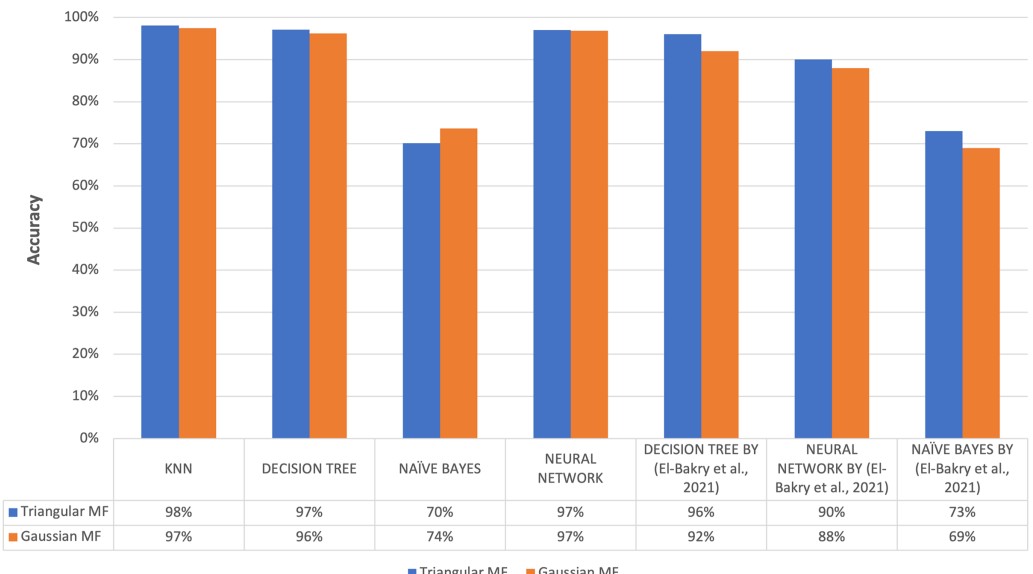

| | KNN | DECISION TREE | NAÏVE BAYES | NEURAL NETWORK | DECISION TREE BY (El-Bakry et al., 2021) | NEURAL NETWORK BY (El-Bakry et al., 2021) | NAÏVE BAYES BY (El-Bakry et al., 2021) |
|---|---|---|---|---|---|---|---|
| ■ Triangular MF | 98% | 97% | 70% | 97% | 96% | 90% | 73% |
| ■ Gaussian MF | 97% | 96% | 74% | 97% | 92% | 88% | 69% |

■ Triangular MF   ■ Gaussian MF

**Figure 15 Comparison of accuracy results of proposed method to previous research.**

the dataset. The process allowed the proposed method to be effective in handling missing values observed in different classification scenarios.

# CONCLUSION

In conclusion, this study proposed a method to combine k-neighbor data and fuzzy membership function for the imputation of missing values. This method produced better accuracy values compared to the conventional imputation methods and triangular fuzzy membership function was observed to have a higher average than Gaussian function. The trend was identified in the 91.72% and 92.07% with a maximum average accuracy at k = 7 recorded for triangular function compared to 89.90% and 90.28% for Gaussian at k = 9. Meanwhile, conventional imputation methods achieved 82.24% for mean imputation, 82.23% for median imputation, and an average accuracy of 76.02% for the original data. This showed that the proposed method was effective in overcoming the problem of missing values in different classification methods and achieved a maximum accuracy of 98.11% with triangular membership function when evaluated using KNN classification method.

## Funding
The authors received no funding for this work.

## Competing Interests
The authors declare that they have no competing interests.

## Author Contributions

- Kurnia Muludi conceived and designed the experiments, performed the experiments, analyzed the data, performed the computation work, prepared figures and/or tables, and approved the final draft.
- Revita Setianingsih conceived and designed the experiments, performed the experiments, analyzed the data, performed the computation work, prepared figures and/or tables, and approved the final draft.
- Ridho Sholehurrohman conceived and designed the experiments, authored or reviewed drafts of the article, and approved the final draft.
- Akmal Junaidi performed the computation work, authored or reviewed drafts of the article, and approved the final draft.

## Data Availability

The code is available at Zenodo: Kurnia Muludi. (2024). kurniadj/missingdata: Initial (Initial). Zenodo. https://doi.org/10.5281/zenodo.10688355.

The hepatitis dataset is available at: Hepatitis. (1988). UCI Machine Learning Repository. https://doi.org/10.24432/C5Q59J.

The ozone level detection dataset (onehr) is available at: Zhang,Kun, Fan,Wei, and Yuan,XiaoJing. (2008). Ozone Level Detection. UCI Machine Learning Repository. https://doi.org/10.24432/C5NG6W.

The blood transfusion service center dataset is available at: Yeh,I-Cheng. (2008). Blood Transfusion Service Center. UCI Machine Learning Repository. https://doi.org/10.24432/C5GS39.

The Parkinson speech dataset is available at: Kursun,Olcay, Sakar,Betul, Isenkul,M., Sakar,C., Sertbas,Ahmet, and Gurgen,Fikret. (2014). Parkinson Speech Dataset with Multiple Types of Sound Recordings. UCI Machine Learning Repository. https://doi.org/10.24432/C5NC8M.

The audit risk dataset is available at: Hooda,Nishtha. (2018). Audit Data. UCI Machine Learning Repository. https://doi.org/10.24432/C5930Q.

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
