# Peer review of "Exploiting nearest neighbor data and fuzzy membership function to address missing values in classification"

_PeerJ Computer Science, doi:10.7717/peerj-cs.1968_

## Round 0.1 · original submission · Major Revisions

This paper introduces a method that combines k-neighbor data and a fuzzy membership function for the imputation of missing values. Since there are some related papers in existing literature, this paper lacks some comparison to show some advantages of the improved algorithm/method.

·

Basic reporting

The manuscript presents a simple, yet effective method to improve quality of imputation while handling datasets with missing values.

In my opinion, the text is clear and didactic enough and I have no remarks upon its presentation, organization, and clearness of ideas.

The references are relevant and recent. I suggest giving more details of the state-of-the-art in imputation methods, with further comments in the literature review section.

Experimental design

The methods are described with good detail. The flowchart included in Figure 1 and the example that shows step-by-step how the missing values are calculated using k-neighbors make the workflow very clear and facilitate its reproducibility.

The results show that the proposed method, combining k-neighbors and fuzzy sets, can improve the quality of imputation.

However, I would suggest that the authors explain with more detail the results shown in Figures 6 to 13, commenting those individually.

Validity of the findings

I consider that the findings are relevant and the conclusions are well stated.
The objectives are clear and were fullfilled.

Reviewer 2 ·

Basic reporting

1. The content of this paper need to be significantly improved before publication
2.Many grammar errors occurred in this paper. Please ask a fluent English speaker to help you to correct the grammar errors occurred in your paper.
3. Modify the reference section.

Experimental design

In my opinion the experimental structure of the paper is acceptable.

Validity of the findings

1.More discussion on real-world application problems in the literature has not been addressed.
2. Although future works are proposed, the limitations of the present work remains unclear (the research assumptions will help provide clarity to the limitations of the work). The content of this paper need to be significantly improved before publication .

Additional comments

The paper may be accepted after minor revision.

---

## Round 0.2 · accepted · Accept

Based on the two reviewers' comments, this article can be accepted for publication.

·

Basic reporting

This reviewed manuscript presents a significative improvement in the use of professional English.
The authors also updated the literature review according to the suggestions.

Experimental design

The suggestions to provide greater detail in the examples were accepted.

Validity of the findings

I consider that the findings are relevant and the conclusions are well stated.
The objectives are clear and were fullfilled.

Additional comments

I reinforce the comments made in the original version regarding the quality of the manuscript.
For this review, I consider that the reviewers' suggestions were implemented and, therefore, it is apt to be published.

Reviewer 2 ·

Basic reporting

no comment

Experimental design

no comment

Validity of the findings

no comment

Additional comments

All the queries are answered.